# Association between subjective tinnitus and cognitive performance: protocol for systematic review and meta-analysis

Nathan A Clarke,[1,2,3] Michael A Akeroyd,[1,3] Helen Henshaw,[2] Derek J Hoare[2]

[1]Medical Research Council Institute of Hearing Research, School of Medicine, The University of Nottingham, University Park, Nottingham, UK
[2]National Institute of Health Research Nottingham Biomedical Research Centre, Nottingham, UK
[3]Hearing Sciences, Division of Clinical Neurosciences, School of Medicine, The University of Nottingham, University Park, Nottingham, UK

**Correspondence to**
Mr Nathan A Clarke;
nathan.clarke@nottingham.ac.uk

## ABSTRACT

**Introduction** Subjective tinnitus is very common and has a number of comorbid associations including depression, sleep disturbance and concentration difficulties. Concentration difficulties may be observable in people with tinnitus through poorer behavioural performance in tasks thought to measure specific cognitive domains such as attention and memory (ie, cognitive performance). Several reviews have discussed the association between tinnitus and cognition; however, none to date have investigated the association between tinnitus and cognitive performance through meta-analysis with reference to an established theoretical taxonomy. Furthermore, there has been little overlap between sets of studies that have been included in previous reviews, potentially contributing to the typically mixed findings that are reported.

**Methods and analysis** This systematic review aims to comprehensively review the literature using an established theoretical taxonomy and quantitatively synthesise relevant data to determine associations between subjective tinnitus and cognitive performance. Methods are reported according to Preferred Reporting Items for Systematic Reviews and Meta-Analyses Protocols. All study designs will be eligible for inclusion with no date restrictions on searches. Studies eligible for inclusion must contain adult participants (≥18 years) with subjective tinnitus and a behavioural measure of cognitive performance. Meta-analysis will be reported via correlation for the association between tinnitus and cognitive performance.

**Ethics and dissemination** No ethical issues are foreseen. Findings will be reported in a student thesis, at national and international , ear, nose and throat/audiology conferences and by peer-reviewed publication.

**PROSPERO registration number** CRD42018085528.

## INTRODUCTION

Tinnitus refers to the common experience of sound in the ears or head in the absence of an external source. It is commonly considered a symptom of damage within the auditory system.[1] Objective tinnitus involves sound with a known, non-central aetiology such as vascular abnormalities; it may be detected by an observer through auscultation. Objective tinnitus may be treated once the source of the aetiology has been identified and is therefore not of primary interest within this review.[2]

### Strengths and limitations of this study

► This systematic review and meta-analysis protocol poses a clearly formulated research question and methodology to investigate a common clinical complaint of patients with tinnitus; peer-reviewed evidence to date will be synthesised.

► This protocol details a comprehensive quantitative synthesis with inclusion of potential a priori moderator variables.

► Synthesis will be clearly structured according to an established cognitive theoretical framework.

► Grey literature and dissertation abstracts will not be included.

Subjective tinnitus (hereafter discussed but simply referred to as 'tinnitus') involves sound of unknown aetiology. Most individuals who experience tinnitus do not find it bothersome but a significant proportion are disturbed by it, often reporting a variety of adverse comorbid associations including anxiety, depression, disturbed sleep or concentration difficulties.[3–5] Concentration difficulties can be conceptualised as failures of cognitive performance expressed behaviourally (sometimes called objective cognition[6]) in various domains such as attention and memory.[7 8] Previous research has implicated tinnitus as negatively impacting cognitive performance in domains including executive functions, attention and working memory.[9–11] Furthermore, a link between subjective perception of cognitive performance, or subjective cognitive complaints (SCC), has also been suggested.[12] Investigating the potential impact of tinnitus on cognitive performance is further complicated by the strong associations that tinnitus shares with depression and anxiety, which can independently negatively impact cognitive performance.[5 13 14]

Several reviews have explored the relationship between tinnitus and cognition generally.[12 15–17] We note that all of these are narrative reviews; no review has quantitatively

**Table 1** Overview of previous reviews investigating cognition and tinnitus

| Study | Databases searched | Example search strategy provided | Number of records retrieved during database search | Number of studies included in review | Synthesis of association between cognition and tinnitus |
|---|---|---|---|---|---|
| Trevis et al[17] | PsycINFO, MedLine | (1) tinnitus AND psych* (all fields), (2) tinnitus AND mood (all fields), (3) tinnitus AND depress* (all fields), (4) tinnitus AND anx* OR stress (all fields) | 725 | 64 | Meta-analysis of 'psychological functioning' in tinnitus participants. Narrative review of association between 'cognitive functioning' (n=16); that is, behavioural cognitive task performance and chronic tinnitus, awareness, and severity. |
| Mohamad et al[12] | PubMed | ((((tinnitus(Title)) AND cogniti*(Title))) OR ((tinnitus(Title)) AND attention(Title))) OR ((tinnitus(Title)) AND memory(Title)) | 65 | 9 | Narrative review of 'behavioural research' addressing the impact of tinnitus and its severity of various aspects of cognitive performance in domains of working memory and attention. |
| Tegg-Quinn et al[16] | PubMed, MedLine, CINAHL, Scopus, EMBASE | (tinnitus) and (cognition OR memory OR attention OR concentration OR cognitive function OR mental activity) NOT (infant OR child OR adolescent OR paediatric OR animal OR balance OR hyperacusis OR implant OR pharmaceutical OR drugs) | 2236 | 18 | Narrative review of behavioural cognitive tasks, electrophysiological correlates of cognition and self-reported cognitive function measures. |
| Andersson and McKenna[15] | MedLine and Psychological Abstracts | Not reported | Not reported | Not reported | Narrative review of 'cognitive deficits' (ie, behavioural cognitive tasks), 'cognitive bias' and 'conscious appraisal of tinnitus'. |

CINAHL; Cumulative Index to Nursing and Allied Health Literature.

synthesised the literature specifically concerning tinnitus and behaviourally measured cognitive performance. An overview of previous reviews and their methodologies is provided in table 1.

Andersson and McKenna[15] were the first to review the relationship between cognition and tinnitus, detailing three separate but related lines of cognitive research. The strands of research included neuropsychological studies involving attention, cognitive bias (concerning selective attention and memory) and appraisal (ie, conscious recollection) of tinnitus. Tegg-Quinn et al[16] performed a systematic review of all studies pertaining to the impact of tinnitus on cognition in adults. The review described studies that included behavioural, electrophysiological and SCC measures. The authors concluded that tinnitus impairs cognition by adversely impacting the executive control of attention. Mohamad et al[12] performed a narrative review of the behavioural evidence concerning the consequences of tinnitus and its severity on cognition. They concluded that there was suggestive evidence for tinnitus being associated with poorer performance in behavioural tasks attempting to measure executive attention, selective attention and working memory. This review also examined the proposed relationship between cognitive performance and SCC in individuals with tinnitus. They reported that their data were insufficient to form conclusions and recommended further investigation of the relationship. Trevis et al[17] performed a systematic review and meta-analysis of psychological functioning in chronic tinnitus. The authors predominantly investigated the presence and severity of tinnitus in relation to emotional well-being through meta-analysis, while cognitive function (ie, cognitive performance) was described through

a narrative review. To summarise, the collective conclusions of these reviews describe mixed evidence in support of the hypothesis that tinnitus adversely impacts cognitive performance and individually included insufficient data to form conclusions regarding associations between cognitive performance and SCC in individuals with tinnitus. Several distinct cognitive functions have been implicated in this hypothesis. Previous studies have suggested that structures relating to auditory attention and efferent structures within the subcallosal region are mechanistically involved in the adverse impacts of tinnitus on cognitive performance.[16] Functional disruption to large-scale neurocognitive networks has also been suggested as a mechanism[17 18]; specifically, a hypoactive cognitive control network and hyperactive 'default mode' or 'task-negative' network.

Of the recent reviews that discuss cognitive performance via behavioural measures in tinnitus participants, there is a notable lack of overlap in the studies that met criteria for inclusion in the final reviews; for example, Mohamad et al[10] reviewed 9 studies and Trevis et al[17] reviewed a total of 64 studies (with 16 of these concerning cognitive performance); however, only three studies were included in both reviews. Therefore, previous work has essentially investigated the association between tinnitus and cognitive performance with different data sets. Schultz et al[19] recently reviewed the evidence for tinnitus impacting neurocognitive profiles following traumatic brain injury. They discuss cognitive performance through selective discussion of aforementioned reviews—except Andersson and Mckenna[15]—and subsequent implications within a medicolegal context. The authors highlight the current lack of and need for empirical investigation of the association between tinnitus and cognitive performance through meta-analysis. Like any statistical technique, meta-analysis is only as robust as the data that are inputted. It is, therefore, essential to include as much relevant data as possible—through a comprehensive search strategy—to ensure that conclusions are based on all of the available evidence concerning the association of tinnitus with specific cognitive domains.

Tinnitus is a symptom of heterogeneous and often unknown aetiologies. It is, therefore, inherently difficult to define and specify within the context of a systematic review. Different inclusion criteria and working definitions of tinnitus are likely to significantly influence the records included within a review. For example, Trevis et al[17] defined 'chronic tinnitus' as participants who had experienced tinnitus for at least 1 month. An alternative approach would be to not attempt to temporally specify a population, but rather investigate this variable through further quantitative analysis if feasible. With regards to domains of cognitive performance, the aforementioned reviews have implicated tinnitus as impacting executive attention, although the evidence is not conclusive: additional domains of cognitive performance are also potentially associated with tinnitus, including selective

attention, working memory and processing speed.[12 15–17] A promising approach to foster empirically valid insights into any association between cognition and tinnitus is through evaluating and categorising tests of cognitive performance according to the theoretical constructs that they are thought to measure.[20] Webb et al[21] describe a cross-disciplinary taxonomy for categorising cognitive performance measures (Cattell-Horn-Carroll-Miyake or CHC-M)). This features combined CHC and Miyake theoretical elements[22 23] and includes a comprehensive taxonomical categorisation of cognitive tasks. CHC-M taxonomy will be used to organise synthesis when investigating the association between tinnitus and cognitive performance. This approach has several benefits: it is informed by the CHC 'three strata' model of cognition, which has been empirically validated through decades of research; it incorporates executive functions, a cognitive construct of particular clinical interest, facilitating translation to the clinical domain; utilisation of a pre-existing taxonomy minimises author bias (as outcome measures are not being subjectively assigned to domains of cognitive performance by authors) and enables comparison compared with 'categorisation as usual'; finally, the taxonomy provides a clear framework around which to structure synthesis of results. Associations between categorised measures of cognitive performance and tinnitus may then be subjected to meta-analysis of specific cognitive domains in order to understand which may beassociated with tinnitus, as well as the best estimate of any such effect. Given the suggestive nature of the evidence provided in previous reviews, we can hypothesise that there will be negative associations between tinnitus severity and cognitive performance in the broad stratum domains of executive functions, processing speed and general short-term memory. To summarise, although several authors have reviewed the association between tinnitus and cognitive performance assessed through behavioural measures, these have been via narrative syntheses. They have discussed different sets of studies, derived from different search strategies in the absence of a unifying taxonomy. A comprehensive, quantitative investigation of the association between tinnitus and cognitive performance, building on earlier efforts in this field and exploring the underlying theoretical domains of cognition involved, is therefore both necessary and timely.

The primary aim of this work is to comprehensively review the literature and synthesise relevant data to determine the associations between tinnitus and cognitive task performance in adults. If possible, a secondary examination of patient characteristics (eg, age or gender) or commonly used patient-reported outcomes (eg, depression or anxiety) and their influence on any association between tinnitus and cognitive performance will also be conducted.

## METHODS AND ANALYSIS

The methodology of this review is reported in accordance with the Preferred Reporting Items for Systematic Reviews and Meta-Analysis Protocols (PRISMA-P) checklist.[24] Specified roles of named authors are identified throughout the review protocol.

## ELIGIBILITY CRITERIA

Only published or in-press, peer-reviewed journal articles will be considered. Articles that are not written in English or have no English language translation available will be excluded as the review team does not have resource available to support translation. No date restriction will be applied.

Review inclusion criteria are specified according to Participant, Intervention (or Interest), Comparator, Outcome and Setting characteristics.

### Participants

Studies including adults (≥18 years) with tinnitus. Studies that include both children (<18 years) and adults will be excluded, unless the adult data are reported separately.

### Intervention/Interest

Tinnitus (via self-report, Tinnitus Severity Scale, item or established tinnitus questionnaires).

### Comparator

A minimum of one established measure of cognitive performance (behavioural or self-report).

### Outcome measures

An association between tinnitus and cognitive performance. Where available, data for associations between tinnitus and additional potential moderator variables will be extracted, such as measures of anxiety or depression.

### Study design

Cross-sectional, longitudinal, experimental, quasi-experimental and observational study designs will be included (only baseline data will be extracted where multiple time-point measurements are made).

## PATIENT AND PUBLIC INVOLVEMENT

There was no patient or public involvement in the development of this manuscript.

## INFORMATION SOURCES AND SEARCH STRATEGY

A systematic search strategy will be employed to identify completed, peer-reviewed journal articles from the following bibliographic databases: MEDLINE (via Ovid SP), EMBASE (via Ovid SP), PsycINFO (via Ovid SP), ASSIA (via ProQuest),Cumulative Index to Nursing and Allied Health Literature or CINAHL(via EBSCO Host), Scopus, PubMed and Web of Science (Science and Social Science Citation Index). Initial searches were performed in February 2018. Update searches will be conducted shortly before manuscript submission.

The search terms used in this systematic review were identified using free text, controlled vocabularies (ie, Medical Subject Headings—MeSH and CINAHL Headings), literature review, opinion of authors and scrutiny of test search results. The following search strategy will be used for PubMed, which will then be adapted for other databases to be searched:

('tinnitus'(MeSH) OR 'tinnitus'(tiab) OR 'phantom sound*"(tiab) OR 'ringing'(tiab) OR 'buzzing'(tiab)) AND ('cognition'(MeSH) OR 'cogniti*"(tiab) OR 'memory'(tiab) or 'attention*"(tiab) OR 'executive'(tiab))

### Data management

NAC will be responsible for data management. Covidence online systematic review software (https://www.covidence.org) will be used for article screening and data management throughout the review. All articles identified through the search process will be recorded digitally and will be trackable through the data screening and extraction processes. Articles excluded at the full text screening stage will have justification for exclusion noted. Included articles will be assigned a unique study identification code, enabling it to be linked to its corresponding full text and data collection sheet.

### Article selection process

NAC, DJH, and MAA will independently screen titles and abstracts of records retrieved from searches such that all records are independently screened by two reviewers. Records that meet the specified inclusion criteria will then be taken forward to full-text screening, as well as records where there is too little information available to make a decision to exclude. All full texts selected will be independently screened by two reviewers who will resolve any discrepancies in which records are included. Where discrepancies for inclusion are not resolved by the two reviewers, a third reviewer will adjudicate. If necessary, study authors will be contacted to request additional information that may help ascertain suitability for inclusion.

### Data extraction process

Data extraction will be performed using a customised form. Relevant guidance notes will be created and disseminated to review team members by NAC prior to commencement. The data form and guidance notes will be piloted by NAC and DJH. All included records will be subject to data extraction. Data from each included record will be extracted independently by two reviewers and the results compared. Any disagreements arising will be resolved through discussion or the involvement of a third reviewer.

### Data items

The data collection form will include various fields corresponding to study type, population, reported effect size and any other relevant study findings. Study authors will be contacted and if there are relevant missing data.

This will entail one email , with instances of no response being reported as such. Data will be approximated from figures for instances where it can only be estimated, using software such as WebPlotDigitizer (http://arohatgi. info/WebPlotDigitizer/app/). Disagreements regarding numerical data extracted from figures will discussed by investigators and/or resolved by averaging.

### Risk of bias in individual studies

NAC, HH, and DJH (two reviewers per record) will independently assess the risk of bias using the Weight of Evidence (WoE) framework, which allows appraisal of study criteria that is tailored to the review question.[25 26] The WoE framework will be customised and used to evaluate *Methodological Quality* (WoE A), *Methodological Relevance* (WoE B) and *Topic Relevance* (WoE C). A rating of low, medium or high will be assigned for each category in accordance with WoE framework criteria. Disagreements regarding bias appraisal will be resolved through discussion or the involvement of a third reviewer.

### Data synthesis

The effect of interest is the association between tinnitus and measures of cognitive performance expressed as correlation. Where possible, results not expressed as correlations (eg, mean differences between groups) will be calculated as appropriate correlations, such as point–biserial or polyserial correlation coefficients.[27] Where possible,bias corrections will be applied to derived correlation coefficients prior to pooling.[28] Missing effect sizes will be calculated from reported test statistics such as SD or t values for records where the relevant information is available.[27] Narrative synthesis will be undertaken for records where appropriate effect sizes cannot be obtained. If SCC measures assess comparable constructs, a separate meta-analysis will be undertaken for the association between tinnitus and SCC. If not, these studies will be synthesised narratively. Cognitive performance will be collapsed over 'broad' factors within the 'level 2' stratum to enable meta-analysis of the association between tinnitus and cognitive performance and these domains. If possible, 'narrow' factors within 'level 3' stratum will be meta-analysed; however, if too few records are included to afford this degree of granulation, then the narrow factors will provide a framework for narrative synthesis.

### Assessment of heterogeneity

The ratio of observed variation to within-study variance will be assessed with the Q-statistic, used to test the null hypothesis of homogeneity of effect sizes. The $I^2$ statistic will provide a further index of heterogeneity across studies. If the apparent heterogeneity across studies exceeds 50%, potential causes of heterogeneity will be explored through subgroup analysis. The $\tau^2$ statistic will also be used to assess the amount of heterogeneity where a random-effects model is fitted to the data. A Baujat plot will be used as a graphical means of identifying studies that contribute excessively to any observed heterogeneity

and also provide insight into potential moderating variables that contribute to heterogeneity across studies.[29]

### Subgroup analyses

Potential contributors to heterogeneity across studies will be explored through subgroup analysis (ie, sensitivity analyses, moderator analysis or meta-regression). A priori variables of interest for subgroup analyses will include tinnitus sample characteristics (duration, laterality, intermittency), study quality and variables known to adversely impact cognitive performance, including sample age, presence of hearing impairment, presence of anxiety or depression, reported medication usage and visual acuity. Additional potential moderating variables may be identified after reviewing the literature and will be documented accordingly.

### Assessment of reporting bias

The influence of reporting bias through potentially unpublished results (ie, publication bias) will be explored via funnel plots, rank correlation test and Egger's regression test.

### Ethics and dissemination

No ethical issues are foreseen in this systematic review. Reports will be guided by the PRISMA guidelines.[30] Various dissemination strategies will be employed that will likely include: journal article and PhD thesis (NAC) made available via an institutional repository, results being reported at national and international academic conferences and public and patient engagement (eg, articles written for relevant non-specialist audiences).

### SUMMARY

A protocol is described for a systematic review and meta-analysis to determine a best estimate of the association between tinnitus and cognitive performance in adults. The relationship between tinnitus, cognitive performance and SSC will also be examined. To date, no review has comprehensively explored the veracity of an association between tinnitus and cognitive performance through application of quantitative analyses of all available peer-reviewed data. The outlined approach will facilitate an understanding of the potential impact of tinnitus on cognitive performance, underpinned by relevant cognitive theory. An increased understanding of the relationship between tinnitus and cognitive performance will eventually improve tinnitus subtyping and inform therapeutic methods[31–34]; for example, it may be possible to deliver cognitive training paradigms in a targeted manner.

**Acknowledgements** Thanks to Oliver Zobay for statistical advice.

**Contributors** NAC led on the development of all sections of the review protocol. DJH, HH and MAA developed and provided feedback for all sections of the review protocol and approved the final manuscript.

**Funding** NAC is supported by a Medical Research Council studentship. MAA is supported by the Medical Research Council (grant number MC_UU_00010/3). HH

and DJH are funded by the National Institute for Health Research (NIHR) Biomedical Research Centre programme.

**Disclaimer** The views expressed are those of the authors and not necessarily those of the NIHR, the NHS or the Department of Health and Social Care.

**Competing interests** None declared.

**Patient consent** Not required.

**Provenance and peer review** Not commissioned; externally peer reviewed.

**Data sharing statement** There are no data currently available.

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
