## [Reviewer comments · BMJ Open]

ARTICLE DETAILS

TITLE (PROVISIONAL)	The association between subjective tinnitus and cognitive performance: protocol for systematic review and meta-analysis
AUTHORS	Clarke, Nathan; Akeroyd, Michael; Henshaw, Helen; Hoare, Derek

VERSION 1 – REVIEW

REVIEWER	Yuchen Chen Nanjing Medical University, China
REVIEW RETURNED	18-May-2018

GENERAL COMMENTS	The authors mainly discussed the associations between subjective tinnitus and cognitive performance and made a protocol by including some narrative reviews. The paper is well written and conducted with proper methods for meta-analysis. However, the number of included reviews is too small (only 4), which make the results not convincing: (1) The biggest problem is the small number of included studies. The authors should check thoroughly the studies published in 2018 and include more studies or reviews in the future. (2) Aims of the study are general with no specific hypotheses. Please elaborate something about the aims as well as the hypotheses. (3) The relationship between chronic tinnitus and cognitive impairment is still unclear. Moreover, the cognitive impairment may contain different cognitive domains affected in different brain regions, including memory, attention, working memory, etc. The authors can discuss the association by selecting one of the cognitive domains. (4) There are several typos and repetitions in the references. Please check it thoroughly.
---

REVIEWER	Vinaya Manchaiah Lamar University, USA
REVIEW RETURNED	19-May-2018

GENERAL COMMENTS	The manuscript provides an excellent protocol for systematic review to examine the association between subjective tinnitus and cognitive performance. The manuscript is generally well written and has appropriate methodology. However, I have few minor comments which can help readability. " The authors make good case for the review by highlighting the limitations of previous reviews. However, not much information presented about effects of tinnitus on cognitive performance. There are number of qualitative and quantitative studies indicating "attention and concentration" and other cognitive aspects are affected in large proportions of tinnitus sufferers. This aspect should
---

	be strengthened in the introduction. In addition, authors should also make effort to mechanisms that may be involved in this. I believe this will help make better argument for review and motivate the study aim. " In data items section, authors indicate that they contact authors of original publications for some missing information relevant to this literature. Some information should be provided on this process (any reminders, how they would contact, what do they do if authors don't respond, etc). " Risk of bias section should have reference to CAPS case control checklist instead of just providing the web link. Refer to BMJ Open reference format. " As there are tracked changes in the manuscript, it appears authors may have submitted the working draft instead of the final version? I suggest highlight any changes made in future to current submitted version.
--	--

REVIEWER	Nikki Hill Pennsylvania State University College of Nursing, United States
REVIEW RETURNED	20-May-2018

GENERAL COMMENTS	This manuscript describes the protocol for a systematic review and meta-analysis of the associations between subjective tinnitus and cognitive performance in adults. Although previous reviews have covered this topic, they have not done so comprehensively (the full body of extant literature) nor quantitatively synthesized via meta-analysis. The authors will address this gap as well as explore patient characteristics and patient-reported outcomes as potential moderators of the associations between tinnitus and cognitive performance. This also extends the scope of this review beyond previous work in the field, and is particularly important for enhancing clinical utility of the review's findings. The protocol is logically organized and has attended to most of the necessary details; however, there are several areas that should be expanded and/or clarified. Most of these are related to the concepts of interest, both conceptual and operational definitions, as well as a need for more justification regarding eligibility criteria and assessment of the strength of the evidence. These recommendations are detailed below. Conceptual and operational definitions: Subjective tinnitus should be distinguished from objective tinnitus when defined, and further clarified as to why objective tinnitus is excluded. There is inconsistent use of the terms "tinnitus", "subjective tinnitus", "self-reported tinnitus" throughout the manuscript. I would recommend clearly defining the concept of interest early, and then it can be referred to as "tinnitus" throughout the rest of the manuscript. Similarly, "self-reported subjective tinnitus" seems redundant (in the purpose statement, abstract, etc.); if not, please clarify as to how this term is meaningful in the current review. It is likely that "subjective tinnitus" is the concept of interest, and it is measured via self-report, but this should be clarified. Cognitive performance is described inconsistently throughout the protocol, including in the study eligibility criteria, and this negatively impacts the protocol overall. It is unclear what is meant by a "behavioral measure of cognitive performance." It is likely this refers to objective cognition, and the statement of the review aim is clear as it states, "cognitive task performance." However, the article
---

eligibility criteria states that cognitive performance measures include "behavioral or self-report" measures. As stated, this proposes that both objective and subjective measures of cognition would be included, which covers two concepts that must be addressed individually in addition to their areas of overlap. Further, if self-reported cognition is to be examined in this review and meta-analysis, then the introduction, background, synthesis, and analytic plan should be adjusted accordingly. The authors might start by consulting the literature on subjective and objective cognition such as Burmester, Leathem, & Merrick (2016): "Subjective cognitive complaints and objective cognitive function in aging: A systematic review and meta-analysis of recent cross-sectional findings," as well as the literature examining subjective cognition as a predictor of objective cognition over time. Furthermore, cognition is inconsistently defined throughout the manuscript (e.g., within the introduction), and it is unclear how specific cognitive domains will be examined in the narrative review and meta-analysis. There is a statement that this investigation will explore the "underlying theoretical domains of cognition involved," which would be a particular strength of the planned review, but the methods described in the rest of the manuscript do not articulate how this will be achieved.

Eligibility criteria:

In addition to the recommendations above, there is a need to justify the inclusion of "any type of study design" in the review. It is unclear how experimental or quasi-experimental studies would inform the review aim, for example. In such cases, baseline data may be informative, but the protocol should explicitly address these issues.

The inclusion of potential moderator variables is a strength of the proposed review. It is unclear at present, however, whether potential moderators will be included based on a priori determination by the authors (specific patient characteristics and patient-reported outcomes that are thought to influence the associations of interest) or whether these will be determined once the literature review has been conducted, and therefore based on what the identified articles included as moderators. These methods should be fully described and justified.

Search strategy: Please expand on who provided expert opinion on the search terms used. Further, the description of the search states that it will be performed during a date range that has already passed.

Risk of bias and evaluation of study quality: It is unclear why the CASP Case Control Checklist is to be used when any type of study is to be included in the review. Further, a complete description of the methods to evaluate study quality is needed.

Data synthesis and analyses: Additional detail is needed regarding the approach to evidence synthesis in general, and in particular with regard to the different cognitive domains of interest. Particular attention should be paid to expanding details in the protocol that fall under the "Data" section of the PRISMA checklist.

Strengths and limitations: The two bullet points under the abstract pertaining to the study strengths and limitations could be expanded to highlight the unique contribution this review will make to the knowledge base in this area. For example, a comprehensive

	quantitative synthesis and inclusion of potential moderators.
REVIEWER	Sarah Granberg School of Health Sciences, Örebro University, Sweden
REVIEW RETURNED	24-May-2018
GENERAL COMMENTS	Thank you for the opportunity to review the current protocol. It is clear and concise and well written. The authors have argued well for the study rationales. I have no comments to the manuscript. I do not consider myself as an expert of the presented statistical analyses, however the authors have consulted a statistical expert (acknowledgements), which I consider to be a strength in the protocol.

VERSION 1 – AUTHOR RESPONSE

Dear reviewing panel,

Thank you for your comments concerning the ‘The association between tinnitus and cognitive performance: protocol for systematic review and meta-analysis’. We are very grateful for your helpful and constructive comments on the manuscript. We have addressed all the comments and implemented all changes that were proposed. We provide a point-by point response below with new manuscript text highlighted in yellow in both the revised manuscript and in reviewer responses. We hope that this protocol is now suitable for publication in BMJ Open.

Yours sincerely,

Nathan A. Clarke, Michael A. Akeroyd, Helen Henshaw and Derek J. Hoare.

Reviewer 1

1.

The biggest problem is the small number of included studies. The authors should check thoroughly the studies published in 2018 and include more studies or reviews in the future.

We performed a search for 2018 publications and found one additional review (Schultz et al. 2018) relative to the aims of this review and meta-analysis. Reference to this review and relevance to protocol have been added: Schultz et al. [19] recently reviewed the evidence for tinnitus impacting neurocognitive profiles following traumatic brain injury. They discuss cognitive performance through selective discussion of aforementioned reviews - except Andersson and Mckenna [15] – and subsequent implications within a medicolegal context. The authors highlight the current lack of and need for empirical investigation of the association between tinnitus and cognitive performance through meta-analysis

2.

Aims of the study are general with no specific hypotheses. Please elaborate something about the aims as well as the hypotheses

We have revised information concerning the aims and hypotheses to be more explicit. The text now reads: Given the suggestive nature of the evidence provided in previous reviews, we can hypothesise that there will be negative associations between tinnitus severity and cognitive performance in the broad stratum domains of executive functions, processing speed, and general short-term memory.

3.

The relationship between chronic tinnitus and cognitive impairment is still unclear. Moreover, the cognitive impairment may contain different cognitive domains affected in different brain regions, including memory, attention, working memory, etc. The authors can discuss the association by selecting one of the cognitive domains

The specification of the CHC-M framework will now facilitate focus on the relationship between chronic tinnitus and specific cognitive domains (see response to Reviewer 3, comment 9).

4.

There are several typos and repetitions in the references. Please check it thoroughly.

Reference formatting has been re-checked and amended accordingly.

Reviewer 2

5.

The authors make good case for the review by highlighting the limitations of previous reviews. However, not much information presented about effects of tinnitus on cognitive performance. There are number of qualitative and quantitative studies indicating "attention and concentration" and other cognitive aspects are affected in large proportions of tinnitus sufferers. This aspect should be strengthened in the introduction. In addition, authors should also make effort to mechanisms that may be involved in this. I believe this will help make better argument for review and motivate the study aim

The conclusions of previous reviews concerning the effects of tinnitus on cognitive performance have been added, as well as reference to the potential mechanisms believed to underlie the proposed impact of tinnitus on cognitive performance: To summarise, the collective conclusions of these reviews describe mixed evidence in support of the hypothesis that tinnitus adversely impacts cognitive performance and individually included insufficient data to form conclusions regarding associations between cognitive performance and SCC in individuals with tinnitus. Several distinct cognitive functions have been implicated in this hypothesis. Previous studies have suggested that structures relating to auditory attention and efferent structures within the subcallosal region are mechanistically involved in the adverse impacts of tinnitus on cognitive performance [16]. Functional disruption to large scale neurocognitive networks has also been suggested as a mechanism [17,18] ;

specifically, a hypoactive cognitive control network and hyperactive 'default mode' or 'task-negative' network.

6.

In data items section, authors indicate that they contact authors of original publications for some missing information relevant to this literature. Some information should be provided on this process (any reminders, how they would contact, what do they do if authors don't respond, etc).

This information has now been included: This will entail one email reminder, with instances of no response being reported as such

7.

Risk of bias section should have reference to CAPS case control checklist instead of just providing the web link. Refer to BMJ Open reference format.

See response to Reviewer 3, comment 13

Reviewer 3

8.

Subjective tinnitus should be distinguished from objective tinnitus when defined, and further clarified as to why objective tinnitus is excluded. There is inconsistent use of the terms "tinnitus", "subjective tinnitus", "self-reported tinnitus" throughout the manuscript. I would recommend clearly defining the concept of interest early, and then it can be referred to as "tinnitus" throughout the rest of the manuscript. Similarly, "self-reported subjective tinnitus" seems redundant (in the purpose statement, abstract, etc.); if not, please clarify as to how this term is meaningful in the current review. It is likely that "subjective tinnitus" is the concept of interest, and it is measured via self-report, but this should be clarified.

The distinction between subjective and objective tinnitus and rationale for exclusion is now included, with consistent use of 'tinnitus' thereafter: Tinnitus refers to the common experience of sound in the ears or head in the absence of an external source. It is commonly considered a symptom of damage within the auditory system [1]. Objective tinnitus involves sound with a known, non-central aetiology vascular abnormalities; it may be detected by an observer using auscultation. Objective tinnitus may be treated once the source of the aetiology has been identified and is therefore not of primary interest within this review [2]. Subjective tinnitus (hereafter discussed but simply referred to as 'tinnitus') involves sound of unknown aetiology.

9.

Cognitive performance is described inconsistently throughout the protocol, including in the study eligibility criteria, and this negatively impacts the protocol overall. It is unclear what is meant by a "behavioral measure of cognitive performance." It is likely this refers to objective

cognition, and the statement of the review aim is clear as it states, "cognitive task performance." However, the article eligibility criteria states that cognitive performance measures include "behavioral or self-report" measures. As stated, this proposes that both objective and subjective measures of cognition would be included, which covers two concepts that must be addressed individually in addition to their areas of overlap. Further, if self-reported cognition is to be examined in this review and meta-analysis, then the introduction, background, synthesis, and analytic plan should be adjusted accordingly. The authors might start by consulting the literature on subjective and objective cognition such as Burmester, Leathem, & Merrick (2016): "Subjective cognitive complaints and objective cognitive function in aging: A systematic review and meta-analysis of recent cross-sectional findings," as well as the literature examining subjective cognition as a predictor of objective cognition over time. Furthermore, cognition is inconsistently defined throughout the manuscript (e.g., within the introduction), and it is unclear how specific cognitive domains will be examined in the narrative review and meta-analysis. There is a statement that this investigation will explore the "underlying theoretical domains of cognition involved," which would be a particular strength of the planned review, but the methods described in the rest of the manuscript do not articulate how this will be achieved.

A clarification of cognitive performance (objective cognition) and subjective cognitive complaints (SCC) has been added, a justification for including SCC in the review has been added, Consistent use concerning this now clarified terminology has also been addressed: Concentration difficulties can be conceptualised as failures of cognitive performance expressed behaviourally (sometimes called objective cognition[6]) in various domains such as attention and memory[7,8]. Previous research has implicated tinnitus as negatively impacting cognitive performance in domains including executive functions, attention and working memory [9–11].Furthermore, a link between subjective perception of cognitive performance, or subjective cognitive complaints (SCC), has also been suggested [12]

The synthesis and analytic plan concerning SCC has been added. As no a priori relationship is hypothesised between cognitive performance and SCC in tinnitus participants, the overlap between these two concepts will be addressed through discussion of analytic results and included via narrative synthesis: If SCC measures assess comparable constructs, a separate meta-analysis will be undertaken for the association between tinnitus and SCC. If not, these studies will be synthesised narratively.

The underlying theoretical constructs to be explored will be based around the CHC-M cognitive taxonomy. The justification for using this taxonomy is also detailed: Webb et al. [21] describe a cross-disciplinary taxonomy for categorising cognitive performance measures (CHC-M). It utilises combined Cattell-Horn-Carroll (CHC) and Miyake theoretical elements [22,23] and includes a comprehensive taxonomical categorisation of cognitive tasks. CHC-M taxonomy will be used to organise synthesis when investigating the association between tinnitus and cognitive performance. This approach has several benefits: it is informed by the CHC 'three strata' model of cognition, which has been

empirically validated through decades of research; it incorporates executive functions, a cognitive construct of particular clinical interest, facilitating translation to the clinical domain; utilisation of a pre-existing taxonomy minimises author bias (as outcome measures are not being subjectively assigned to domains of cognitive performance by authors) and allows comparison compared to 'categorisation as usual'; finally, the taxonomy provides a clear framework around which to structure synthesis of results.

10.

In addition to the recommendations above, there is a need to justify the inclusion of "any type of study design" in the review. It is unclear how experimental or quasi-experimental studies would inform the review aim, for example. In such cases, baseline data may be informative, but the protocol should explicitly address these issues.

The inclusion criteria for studies has been amended and inclusion of experimental/quasi-experimental data justified through usage of baseline data: Cross-sectional, longitudinal, experimental, quasi-experimental and observational study designs will be included (only baseline data will be extracted where multiple timepoint measurements are made).

11.

The inclusion of potential moderator variables is a strength of the proposed review. It is unclear at present, however, whether potential moderators will be included based on a priori determination by the authors (specific patient characteristics and patient-reported outcomes that are thought to influence the associations of interest) or whether these will be determined once the literature review has been conducted, and therefore based on what the identified articles included as moderators. These methods should be fully described and justified.

We are using moderators identified both a priori and post-hoc. The a priori moderating variables of interest have been stated. Any post-hoc moderators may be identified from literature review of the included records; and a statement to this effect has been added: *A priori* variables of interest for subgroup analyses will include tinnitus sample characteristics (duration, laterality, intermittency), study quality and variables known to adversely impact cognitive performance, including sample age, presence of hearing impairment, presence of anxiety or depression, reported medication usage and visual acuity. Additional potential moderating variables may be identified after reviewing the literature and will be documented accordingly.

12.

Search strategy: Please expand on who provided expert opinion on the search terms used. Further, the description of the search states that it will be performed during a date range that has already passed.

Expert opinion regarding search terms was that of the manuscript authors. This has now been explicitly stated as such. The correct past-tense for the initial searches that have been undertaken has now been corrected: Initial searches were performed in February 2018. Update searches will be conducted shortly before manuscript submission. The search terms to be used in this systematic review were identified using free text, controlled vocabularies (i.e. Medical Subject Headings - MeSH and CINAHL Headings), literature review, opinion of authors, and scrutiny of test search results.

13.

Risk of bias and evaluation of study quality: It is unclear why the CASP Case Control Checklist is to be used when any type of study is to be included in the review. Further, a complete description of the methods to evaluate study quality is needed.

We have carefully considered these comments and agree that using the CASP Case Control Checklist is not conducive to our aim of including different study types within the review. We will therefore be using the Weight of Evidence (WoE) framework to address critical appraisal of the studies. Using this framework will enable us to assess appraisal of criteria that are relevant to the aims of the review.

14.

Data synthesis and analyses: Additional detail is needed regarding the approach to evidence synthesis in general, and in particular with regard to the different cognitive domains of interest. Particular attention should be paid to expanding details in the protocol that fall under the "Data" section of the PRISMA checklist.

The data synthesis section has been updated with specific analytic plans in relation to the CHC-M taxonomy of cognitive factors: Cognitive performance will be collapsed over 'broad' factors within the level '2' stratum to enable meta-analysis of the association between tinnitus and cognitive performance these domains. If possible, 'narrow' factors within 'level 3' stratum will be meta-analysed, however, if too few records are included to afford this degree of granulation, then the narrow factors will provide a framework for narrative synthesis.

15.

Strengths and limitations: The two bullet points under the abstract pertaining to the study strengths and limitations could be expanded to highlight the unique contribution this review will make the knowledge base in this area. For example, a comprehensive quantitative synthesis and inclusion of potential moderators.

The strengths and limitations have been expanded to include a further strength of the proposed protocol being:

- This systematic review and meta-analysis protocol poses a clearly formulated research question and methodology to investigate a common clinical complaint of tinnitus patients; peer-reviewed evidence to date will be synthesised.
- This protocol details a comprehensive quantitative synthesis and inclusion of potential *a priori* moderator variables
- Synthesis will be clearly structured according to established cognitive theoretical frameworks
- Grey literature and dissertation abstracts will not be included

VERSION 2 – REVIEW

REVIEWER	Nikki L. Hill Penn State University, USA
----------	---

REVIEW RETURNED	08-Jul-2018
GENERAL COMMENTS	The authors have adequately addressed my recommendations in this revised manuscript.